



# Mean and extreme precipitation over European river basins better simulated in a 25km AGCM

Reinhard Schiemann[1], Pier Luigi Vidale[1], Len C. Shaffrey[1], Stephanie J. Johnson[1,3], Malcolm J. Roberts[2], Marie-Estelle Demory[1,4], Matthew S. Mizielinski[2], and Jane Strachan[1,2]

[1]National Centre for Atmospheric Science, Department of Meteorology, University of Reading, United Kingdom
[2]Met Office Hadley Centre, Exeter, United Kingdom
[3]European Centre for Medium-Range Weather Forecasts, Reading, United Kingdom
[4]Center for Space and Habitability, University of Bern, Switzerland

*Correspondence to:* Reinhard Schiemann (r.k.schiemann@reading.ac.uk)

**Abstract.** Limited spatial resolution is one of the factors that may hamper applications of global climate models (GCMs), in particular over Europe with its complex coastline and orography. In this study, the representation of European mean and extreme precipitation is evaluated in simulations with an atmospheric GCM at different resolutions between about 135 km and 25 km grid spacing in the midlatitudes. The continent-wide root-mean-square error in mean precipitation in the 25km model

is about 25% smaller than in the 135km model in winter. Clear improvements are also seen in autumn and spring, whereas the model's sensitivity to resolution is very small in summer. Extreme precipitation is evaluated by estimating generalised extreme value distributions (GEVs) of daily precipitation aggregated over river basins whose surface area is greater than 50000 km². GEV location and scale parameters are measures of the typical magnitude and of the interannual variability of extremes, respectively. Median model biases in both these parameters are around 10% in summer and around 20% in the other

seasons. For some river basins, however, these biases can be much larger and take values between 50% and 100%. Extreme precipitation is better simulated in the 25km model, especially during autumn when the median GEV parameter biases are more than halved, and in the Northern European Plains from the Loire in the west to the Vistula in the east. A sensitivity experiment is conducted showing that these resolution sensitivities in both mean and extreme precipitation are in many areas primarily due to the increase in resolution of the model orography. The findings of this study illustrate the improved capability of a global

high-resolution model in simulating European mean and extreme precipitation.

## 1  Introduction

There is an obvious requirement for climate models to simulate precipitation in a realistic way if the models are to be applied, for example, in process and impact studies of the hydrological cycle (e.g., Music and Caya, 2007; Middelkoop et al., 2001; Christensen et al., 2004), prediction at different lead times (e.g. Yuan et al., 2015; Arnell and Gosling, 2016; Scoccimarro et al.,

2016), and extreme event attribution (e.g., Stott et al., 2016; Schaller et al., 2016). Due to the wide range of applications, the required realism concerns all aspects of the precipitation distribution in space and time including the probability distribution function of the precipitation time series and its extremes.





Limited grid resolution is one of the factors that may hamper model application. At the turn of the millenium, state-of-the-art Global Climate Models (GCMs) had grid spacings of several hundred kilometers (McAvaney et al., 2001). This resolution is too low for many applications, particularly so over Europe with its complex coastline and orography, and higher-resolution regional climate models (RCMs), driven at their boundaries by GCM or reanalysis data, were used to overcome this limitation. At the time, RCMs' grid spacings were on the order of 50 km (Giorgi et al., 2001).

Model development and the availability of more and more powerful computing and data analysis facilities have changed this situation radically. Multidecadal GCM simulations can now be carried out at grid spacings of about 20 km (Jung et al., 2012; Mizuta et al., 2012; Wehner et al., 2014; Mizielinski et al., 2014; van Haren et al., 2015). This offers the possibility of using a single global physically consistent model in applications that until a few years ago were in the realm of RCM simulations (van der Linden and Mitchell, 2009). GCMs also remain important for providing initial and boundary conditions for RCMs (e.g., Kendon et al., 2010), which now yield km-scale climate simulations (Hohenegger et al., 2008; Kendon et al., 2014; Prein et al., 2015) allowing for convection-permitting simulations crucial for representing in particular sub-daily precipitation extremes (Ban et al., 2015; Kendon et al., 2017) and the soil moisture–precipitation feedback (Hohenegger et al., 2009).

The aims of this study are threefold. Firstly, rapid model development requires continued model evaluation, and we evaluate here the representation of European precipitation in the UPSCALE atmospsheric GCM (AGCM) simulations (Mizielinski et al., 2014), which are still rather exceptional in their combination of model resolution, simulation length, and ensemble size. We evaluate both seasonal mean and extreme precipitation. These evaluation results are to serve as a benchmark for future generations of GCMs with grid spacings on the order of 10 km, as well as for km-scale RCM simulations. The second aim of this study is to determine to what extent the resolution sensitivity in precipitation is due to the sensitivity to resolution of the simulated North Atlantic storm track as shown in previous studies (Zappa et al., 2013; van Haren et al., 2015), and to contrast that with the role of local forcing from the orography at different resolutions. The third aim concerns the methodology used for evaluating extreme precipitation. We combine two approaches used previously: we characterise daily precipitation extremes by fitting extreme value distributions as done, for example, by Zwiers and Kharin (1998), Frei et al. (2006), and Chan et al. (2014). Furthermore, we conduct model evaluation over large (>50000 km$^2$) river basins in Europe. At these scales, evaluation has typically been carried out for RCMs in the past, often coupled to hydrological impact models. These studies have shown that the choice of the driving GCM can be the largest source of uncertainty in the modelling chain (Graham et al., 2007). Due to the recent increase in global model resolution, we can meaningfully evaluate a GCM at such scales in this study. Coupling to impact models is not part of our study, yet we choose to conduct the evaluation for river basins in the hope that our results will prove informative for such future applications.

The manuscript is structured as follows: section 2 describes the climate model and simulations, and the observational reference data. The mean precipitation distribution is evaluated in Sect. 3, and the representation of precipitation extremes over European river basins is evaluated in Sect 4. The roles of the North Atlantic storm track and of the orography are discussed in Sect. 5. We conclude the paper in Sect. 6.





## 2 Methods and data

### 2.1 Model ensemble

The AGCM used here, HadGEM3-GA3.0, is the Global Atmosphere 3.0 configuration of the HadGEM3 family of the Met Office Unified Model (Walters et al., 2011). We use an ensemble of simulations at three different horizontal resolutions: N96

with a grid spacing of 130 km at 50°N, N216 (60 km), and N512 (25 km). There are 85 vertical levels, the same at all three resolutions. These simulations were conducted within the UPSCALE (UK on PRACE: weather-resolving Simulations of Climate for globAL Environmental risk) project and their generation and analysis required exceptionally large computing resources and big data infrastructure (Mizielinski et al., 2014). These resources allowed to produce five simulations at N512 resolution, each lasting 26 years from 1986 to 2011 and using OSTIA sea surface temperature forcing (Donlon et al., 2012).

Lower-resolution simulations with the same forcing were conducted at N216 resolution ($3 \times 26$ years) and at N96 resolution ($5 \times 26$ years). These experiments were designed to test the sensitivity of the simulated climate to horizontal resolution only and therefore retuning at the different resolutions has been kept to a minimum (see Demory et al., 2014; Johnson et al., 2016, for more details on this point). Convection is parameterised at all three resolutions.

We have also conducted a sensitivity experiment to test the role played by the orography boundary conditions as the res-

olution is increased. This experiment was conducted with the HadGEM3-GA6.0 configuration of the Met Office Unified Model (Walters et al., 2017) and is described in Sect. 5.2. HadGEM3-GA6.0 was the closest available model configuration to HadGEM3-GA3.0 at the time the sensitivity experiment was conducted. Differences between GA6.0 and GA3.0 include small adaptations to the semi-implicit semi-Lagrangian dynamical core from "New Dynamics" (Davies et al., 2005) to ENDGame (Even Newer Dynamics for General atmospheric modeling of the environment; Wood et al., 2014) and the new "5A" subgrid

orographic drag parametrisation (Vosper, 2015; Wells, 2015) replacing the previous "4A" scheme (Webster et al., 2003).

### 2.2 Observed precipitation

We evaluate the simulated precipitation against gridded observed precipitation from the E-OBS dataset (Haylock et al., 2008), version 9. As part of the ENSEMBLES project (van der Linden and Mitchell, 2009), this dataset was designed for the evaluation of daily precipitation across Europe in RCMs at similar resolutions to the AGCM simulations used here. This makes E-

OBS the dataset of choice for our purposes and we refer to differences of model precipitation from E-OBS as model *biases*. Nonetheless, gridded precipitation data are subject to uncertainties from both the measurement error of point observations and the limited spatial representativity of gauges (see, e.g., Frei and Schär, 1998, for an overview). A general dry bias, which can be excacerbated in mountainous terrain and for localized extremes, has been reported for E-OBS (Hofstra et al., 2009; Flaounas et al., 2012; Isotta et al., 2015).





**Table 1.** Seasonal mean precipitation in Europe in mm day$^{-1}$ (land area in -12–50°E, 35–72°N including all model grid boxes with a land fraction greater than 0.5). Based on the ensemble spread, approximate 95% confidence intervals are within $\pm$0.06 mm day$^{-1}$ for N96 and N512, and within $\pm$0.1 mm day$^{-1}$ for N216.

|        | DJF       | MAM       | JJA       | SON       |
|--------|-----------|-----------|-----------|-----------|
| E-OBS  | 1.64      | 1.48      | 1.78      | 1.87      |
| N96    | 2.09±0.05 | 2.17±0.05 | 2.15±0.03 | 2.05±0.03 |
| N216   | 2.05±0.10 | 2.19±0.07 | 2.09±0.06 | 2.06±0.05 |
| N512   | 2.08±0.05 | 2.22±0.01 | 2.06±0.02 | 2.03±0.06 |

## 2.3 Extreme value analysis

We evaluate daily extreme precipitation by fitting a Generalized Extreme Value (GEV) distribution to daily precipitation averaged over a number of European river basins (Table S1). We choose basins with surface areas larger than 50000 km$^2$ so that even at N96 resolution the basins will be represented by several model grid boxes. The GEV distribution is defined as in Coles (2001) and is characterized by three parameters referred to as location $\mu$, scale $\sigma$, and shape $\xi$. Illustrations of how these three parameters influence the GEV distribution will be provided in Sect. 4.1 and Fig. S1. We estimate the GEV parameters using the block maxima approach, where each block consists of daily mean precipitation in a river basin throughout one season. The estimation is carried out in a two-step process. First, we estimate $\mu$, $\sigma$, and $\xi$ for each basin and each season using maximum likelihood estimation (e.g., Coles, 2001). We find that this yields plausible spatial variations of $\mu$ and $\sigma$ with typically similar values in neighbouring basins, while there is considerable scatter in $\xi$ with no systematic dependence on the basin location or area. This indicates that the shape parameter cannot be robustly fitted for each basin separately with the available data. We therefore conduct a second estimation step, where we fix the value of $\xi$ to the average value of all basins for each season, and then, also using maximum likelihood estimation, determine the values of $\mu$ and $\sigma$ for each basin. Uncertainty in the fitted GEV parameters is estimated by parametric resampling.

## 3 Mean precipitation

European mean precipitation in the different seasons is shown in Table 1. HadGEM3-GA3.0 precipitation is 10–50% larger than in E-OBS at all three resolutions. This difference likely reflects a wet bias of the model, but the magnitude of this bias is hard to assess because of the known problems associated with estimating area-average precipitation from a network of gauges (see also Sect. 2.2). Despite the fact that no retuning has been performed at the different resolutions, the resolution sensitivity in the Europe-wide mean precipitation is small (<0.1 mm day−1) and not systematic across the seasons.

The observed climatological mean precipitation distribution for winter and summer is shown in Fig. 1a,d. During winter, there is a general continental-scale gradient from higher precipitation in western Europe to lower precipitation in eastern Europe, with pronounced mesoscale variations. Particularly wet regions are west or north-west facing coasts and/or mountains



**Figure 1.** Climatological mean precipitation (mm day$^{-1}$, 1986–2011) according to (a,d) E-OBS, (b,e) N96 and (c,f) N512 for (a–c) December–February and (d–f) June–August. Contour levels are not equidistant to better capture mesoscale variations.

such as the northwest of the Iberian Peninsula, Ireland, Scotland, the Norwegian coast, and an area between the Alps and the North Sea. Moreover, areas of high coastal precipitation, for example in northwest Spain and the west of the British Isles and Norway, are better resolved at N512 resolution. In summer, the continental-scale gradient is from the dry Mediterranean in the south to much wetter conditions in central and northern Europe, especially so over the Alps, British Isles, southern Scandinavia, and an area extending from the Carpathians to the White Sea.

During winter (Fig. 1b,c), the HadGEM3-GA3.0 wet bias can be seen throughout the continent and is particularly apparent in the north and west of the European mainland. Despite this general wet bias, the N512 model can capture some mesoscale variations that are absent or poorly represented in the N96 model. An example is the comparatively dry swath ranging from the southeast of England to the Mediterranean Sea, arguably in the "rainshadow" of the British Isles and western France. In summer (Fig. 1e,f), wet biases over regions of high topography such as the Pyrenees, Scandinavian Mountains, Alps, Carpathians, and



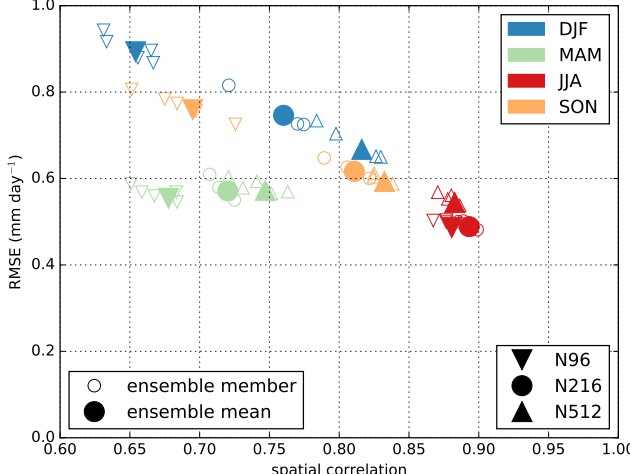

**Figure 2.** Root-mean-square error (RMSE) and spatial correlation between HadGEM3-GA3 precipitation and E-OBS observations in Europe (-14–50°E, 38–70°N). The domain-mean bias is discarded before calculating the RMSE.

the Caucasus, can be seen in the N512 model. There is also a wet bias in both the N96 and N512 models in the north and northeast of Europe. This is consistent with results in Schiemann et al. (2017) (their Figures 7 and 15) showing for this model that there is a mean negative geopotential height bias and an underestimation of summer blocking in the Baltic area.

We proceed with two quantitative evaluations of seasonal mean model precipitation. The root-mean-square error (RMSE)
between the model simulations and E-OBS is plotted in Fig. 2 against the spatial correlation between the same two fields, so that the better the agreement of the model simulation is with E-OBS, the closer will the corresponding entry be to the lower-right corner of the diagram. There is an improvement in the simulated precipitation with resolution in autumn, winter, and spring, and this improvement is significant in the sense that it can be seen in all ensemble members. The improvement in winter is larger than in the transition seasons. In summer, the sensitivity to resolution is very small, the RMSE is slightly larger
in the N512 model than in N96 and N216, arguably due to the higher precipitation over mountainous regions in this model (Fig. 1).

We also conduct scale-dependent evaluation and calculate the Fractions Skill Score (FSS) for different horizontal scales following Roberts and Lean (2008). The FSS is obtained by comparing binary fields, defined in terms of exceedance of a threshold, between model and observation for different sizes of an averaging neighbourhood, i.e. for different horizontal scales.
The FSS takes values between zero and one and typically increases with horizontal scale as shown in Fig. 3 of Roberts and Lean (2008). Here, we use different quantiles of each of the precipitation fields as the exceedance threshold in the FSS calculation so that the FSS approaches unity for large scales and the domain-mean bias (Table 1) is disregarded in this evaluation. In Fig. 3, instead of showing the FSS directly, we show the relative improvement/deterioration of N512 versus N96 calculated in terms of the distance from the FSS = 1 asymptote as $|\text{FSS}_{N512} - \text{FSS}_{N96}|/(1 - \min(\text{FSS}_{N96}, \text{FSS}_{N512}))$. For winter (Fig 3a), we find
an improvement for all quantile thresholds, which is consistent with Figures 1a,b,c and 2. This improvement with resolution is



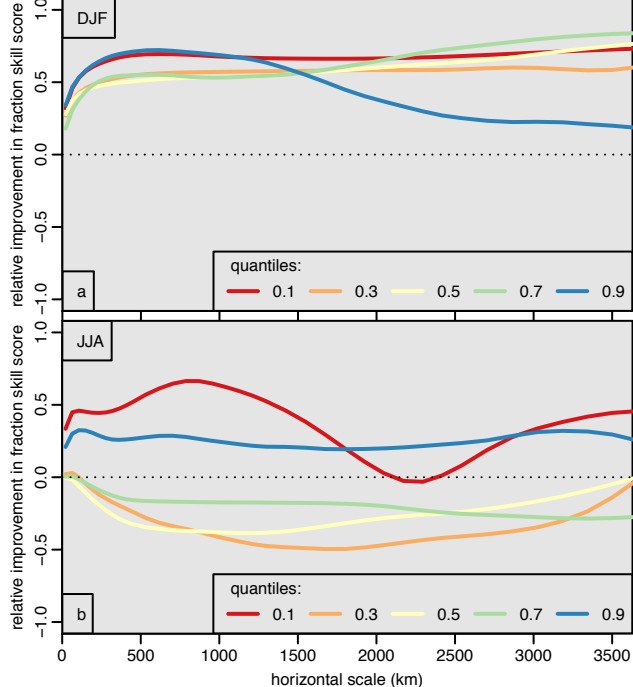

**Figure 3.** Relative improvement in the Fractions Skill Score (FSS) of HadGEM3-GA3.0 N512 over N96 using E-OBS seasonal mean precipitation as reference, for (a) December–February and (b) June–August, and for different quantile thresholds of the spatial precipitation distribution in Europe (-14–50°E, 38–70°N).

seen across all horizontal scales. For summer (Fig 2b), there is no systematic improvement of the mean precipitation field with resolution, also in agreement with the previous analyses.

# 4 Extreme precipitation

## 4.1 Examples

5   In this section, we present results of extreme value analysis and evaluate the amount, frequency, and annual cycle of extreme precipitation in terms of GEV distributions. GEV distributions are conveniently shown in terms of return value plots, also called Gumbel diagrams. The effect of the GEV parameters on the distribution is illustrated for ficticious data in Fig. S1. The larger the values of these parameters, the larger are the precipitation extremes. An increase in the location parameter $\mu$ corresponds to a constant increase in return value for all return times (Fig. S1a). The scale parameter $\sigma$ is associated with the

10   interannual variability of extreme precipitation. The larger $\sigma$, the larger the increase in return value for a given increase in return time (Fig. S1b). The shape parameter $\xi$ determines the behaviour of the tail of the GEV distribution and determines if



the distribution is bounded ($\xi < 0$) or unbounded ($\xi > 0$) for large return times (Fig. S1c). The shape parameter is held constant for all basins as explained in section 2.3 ($\xi = -0.13, -0.05, -0.02, -0.05$ for DJF, MAM, JJA, SON, respectively).

Examples of fitted GEV distributions for three river basins are shown in Fig. 4. Each panel shows GEV distributions for the four seasons alongside the observed precipitation maxima. The top row (Fig 4a–c) shows results for the Loire river basin. The

annual cycle of extreme precipitation is not very pronounced for this basin, and estimated 50 year return values are between about 22 mm day$^{-1}$ (spring) and 26 mm day$^{-1}$ (autumn), although the differences between the seasons are not statistically significant (Fig 4a). The precipitation extremes simulated by the N96 model are very different from those in the observations. The estimates are larger than for E-OBS (50 year return values between about 30 mm day$^{-1}$ for summer and 40 mm day$^{-1}$ for winter) and there is a clear annual cycle with larger extremes during the cold season (Fig. 4b). At N512 resolution, the model

simulated extremes are in closer agreement with E-OBS than at N96 resolution (Fig. 4c): the annual cycle is very small and 50 year return values are between 23 and 29 mm day$^{-1}$. Quantitatively, the reduction of the extreme precipitation biases can be corroborated by comparing the estimated values of the GEV location parameter $\mu$ and scale parameter $\sigma$ between E-OBS, N96, and N512. For all seasons, E-OBS and N512 agree more closely with one another than with N96 (Fig 4a–c). For the Loire basin, the biases in modelled extreme precipitation and their reduction at N512 resolution are consistent with the results seen

for mean precipitation, i.e. a winter wet bias at N96 resolution that is alleviated at N512 resolution (Fig. 1).

Fitted GEV distributions for the Elbe basin are shown in Fig. 4d–f. For this basin, there is a pronounced annual cycle of extremes. The largest extremes occurr during summer with a 50 year return value of about 30 mm day$^{-1}$, while the same return value for winter is only about 17 mm day$^{-1}$ (Fig. 4d). The amplitude of the annual cycle is underestimated in the N96 model. The quantitative agreement with E-OBS is close in summer (50 year return values of about 30 mm day$^{-1}$ in both

datasets), but the simulated winter extremes are larger than the observed ones and the N96 50 year return value is about 25 mm day$^{-1}$ (Fig. 4e). At N512 resolution, the model better captures the annual cycle of extreme precipitation but also somewhat overestimates the 50 year return values, which are about 34 mm day$^{-1}$ in summer and 19 mm day$^{-1}$ in winter. For a shorter return period of 2 years, however, all three datasets are in close agreement on a return value of about 15 mm day$^{-1}$ during summer. This example illustrates how the quantification of a single return value or return period only insufficiently characterises

extreme precipitation, which is why two GEV parameters are used for quantitative model evaluation in this study.

The third basin we discuss is that of the Po river (Fig. 4g–i). In this basin, daily precipitation extremes are considerably larger in autumn than in the other seasons. This behaviour is also captured by HadGEM3-GA3.0 at both resolutions, yet the magnitude of the simulated extremes is much larger than in E-OBS, especially at N512 resolution. The 50 year return value for autumn is about 53 (55, 73) mm day$^{-1}$ in E-OBS (N96, N512). Moreover, and especially during winter, both the N96 and

N512 models overestimate the interannual variability of precipitation extremes as shown by the overestimation of the scale parameter $\sigma$: for return periods of less than 2 years, a number of winter seasons never exceed a precipitation amount of 10 mm day$^{-1}$, which is not seen in E-OBS. On the other hand, winter precipitation extremes are overestimated by our model for return periods greater than 2 years. These results clearly show that the resolution increase from 135 to 25 km is not a panacea for improving the representation of extreme precipitation, and we will summarise results for all of the European basins in the

remainder of section 4 to assess the overall effect of increased resolution on model performance.






**Figure 4.** Examples of fitted GEV distributions for (a–c) the Loire, (d–f) the Elbe, and (g–i) the Po river basins, and for (a,d,g) E-OBS observed precipitation, (b,e,h) the N96 model, and (c,f,i) the N512 model. 95% resampling confidence intervals are shown as shaded areas for winter (DJF) and summer (JJA). Observed maxima (circles) are shown at return periods $\frac{m+1}{m+1-i}$, where $m$ is the number of these maxima and $i = 1, \ldots, m$ is their rank.





Before proceeding with this assessment, we briefly discuss the goodness of the GEV distribution fits, which can be assessed by comparing the observed maxima (open circles) with the parametric GEV fits (solid lines) for the different examples shown in Fig. 4. For most cases, the statistical model fits the observed maxima well, but there are a few discrepancies for larger return periods of more than about 20 years. For example, for the Po and in particular the Elbe basin, a small number of very

heavy precipitation events, which exceed the GEV fit by more than the sampling uncertainty, can be seen during summer for the observations and the N512 model (Fig. 4d,f,g,i). Such discrepancies are partly due to the fact that we choose a constant shape parameter $\xi$ for all basins (see section 2.3). These results illustrate that our parametric approach is valid in general and serves our purpose of characterising the variation of daily extreme precipitation across European river basins, and of evaluating GCM simulated extreme precipitation. Especially for return periods of more than 20 years, however, our estimates of return

values in individual basins/seasons should not be used as the only source to inform impact studies or adaptation and mitigation measures. While our results can provide intial guidance for such applications, they will need to be considered in the context of local expertise, process-based case studies of individual heavy precipitation events, and additional local observations if available.

### 4.2 Winter

The estimates of the location and scale GEV parameters for winter based on E-OBS are shown in Fig. 5a,d. There is a large-scale gradient from high values of $\mu$ and $\sigma$ in the west and southwest of Europe to smaller values in the east and northeast. This geographical variation is similar to that in mean precipitation (Fig. 1a), but there are some interesting differences. For example, comparatively high values of both GEV parameters are seen for some southern European basins (e.g., Po, Guadalquivir) even though the mean precipitation for these basins is not larger than for basins further north. This may indicate fewer but stronger

precipitation events in these basins.

The N96 bias in the GEV parameters is shown in Fig. 5b,e. As can been seen from the predominance of green colours, both the location and scale parameters tend to be overestimated, so the wet bias seen for mean precipitation (section 3) is also found in the extremes. The magnitude of the bias varies strongly between basins and can be as high as about +60% for $\mu$ (Elbe) and about +80% for $\sigma$ (Loire), and the median of the absolute relative bias, i.e. of $\max\left(\frac{\theta_{\text{N96}}}{\theta_{\text{E-OBS}}}, \frac{\theta_{\text{E-OBS}}}{\theta_{\text{N96}}}\right) - 1$, across all basins is 22%

for $\theta = \mu$ and 17% for $\theta = \sigma$ (Table 2). The wet bias is particularly pronounced for the basins in the Northern European Plains, from the Loire in the west to the Vistula in the east, where both $\mu$ and $\sigma$ are overestimated. For some southern European basins (Guadalquivir, Ebro, Po), there are significant negative biases in $\mu$ and significant positive biases in $\sigma$ indicating a difference in the character of the simulated and observed extreme value distribution.

### 4.3 Summer

During summmer, the GEV location parameter $\mu$ takes larger values in the centre, northwest and northeast of Europe than in the southwest (Iberia), southeast and east (Fig. 6a), in rough agreement with the geographical variation of mean precipitation (Fig. 1d). Particularly high values of $\mu$ are seen for basins draining the Alps (e.g., Rhône, Po, Upper Danube) and for the Kuban draining the Caucasus Mountains. The scale parameter $\sigma$ generally follows the geographical distribution of $\mu$, but we







**Figure 5.** Estimated GEV parameters for December–February daily basin-average precipitation, (a–c) location parameter $\mu$ and (d–f) scale parameter $\sigma$, both in mm day$^{-1}$, and for (a,d) observations (E-OBS), (b,e) N96, and (c,f) N512 resolution. Stippling (hatching) shows statistically significant differences between the models and E-OBS (between N512 and N96).

find comparatively larger values of $\sigma$ for drier climates, as can be seen, for example, when comparing the Iberian to the French river basins.







**Figure 6.** As Fig 5 but for summer (June–August).

More so than in winter, there is a dependence of both $\mu$ and $\sigma$ not only on geographical location and local climate, but also on the size of the river basin considered. Larger extremes, i.e. greater values of $\mu$ and $\sigma$, are seen for smaller basins. A case in point are the Dniester and Southern Bug basins compared to the neighbouring larger basins of the Danube and Dnieper. This appears to be due to the nature of summer (convective) precipitation, which is smaller in scale and shorter in duration





**Table 2.** Median across all basins of absolute relative bias (%, see text Sect. 4.2) in $\mu$ and $\sigma$.

|  | $\mu$ | | | $\sigma$ | | |
|  | N96 | N216 | N512 | N96 | N216 | N512 |
|---|---|---|---|---|---|---|
| DJF | 22 | 16 | **16** | 17 | 16 | **16** |
| MAM | 25 | 21 | **20** | 19 | **18** | 19 |
| JJA | **9** | 10 | 10 | **10** | 14 | 12 |
| SON | 18 | 12 | **8** | 17 | 12 | **8** |

**Table 3.** Number of basins with smallest bias in $\mu$ and $\sigma$. For example, the number 4 for $\mu$ in DJF at N96 resolution means that for 4 out of all 33 basins the bias in $\mu$ of the N96 model is smaller than that of both the N216 and N512 models.

|  | $\mu$ | | | $\sigma$ | | |
|  | N96 | N216 | N512 | N96 | N216 | N512 |
|---|---|---|---|---|---|---|
| DJF | 4 | **15** | 14 | 11 | 8 | **14** |
| MAM | 12 | 6 | **15** | **12** | 11 | 10 |
| JJA | **14** | 10 | 9 | **14** | 10 | 9 |
| SON | 4 | 9 | **20** | 7 | 7 | **19** |
| $\Sigma$ | 34 | 40 | **58** | 44 | 36 | **52** |

than frontal winter precipitation. Convective summer rain, though locally intense, may therefore not take large values when averaged over a large river basin over a day.

The biases of $\mu$ and $\sigma$ for the N96 model are shown in Fig. 6b,e. These biases are generally smaller than in winter, and for many basins they are not statistically significant. There is a general tendency for dry biases in the south and wet biases in the north of Europe, especially in $\mu$. The median relative bias across all basin is 9% for $\mu$ and 10% for $\sigma$ (Table 2).

The sensitivity to the resolution increase (Fig. 6c,f) is smaller than in winter, consistent with the results for mean precipitation, and has mixed effects for the biases with respect to E-OBS. At N512, stronger heavy precipitation is seen over the Alps, rather than to the northwest of the Alps over France at N96, leading to weaker extremes and smaller biases for some basins (especially Loire, Seine) and to stronger extremes and a larger bias in particular for the Po basin.

## 4.4 Summary statistics

Two metrics are used to summarise model performance. The first metric is the median absolute relative bias as already introduced and discussed in Sections 4.2 and 4.3. The values of this metric are shown in Table 2. The second metric is obtained by counting for each resolution for how many basins the minimum bias is attained at this resolution, so that a higher count corresponds to a better model performance. The values of this metric are shown in Table 3.



Both metrics agree on the following qualitative results: extreme precipitation is overall better represented as resolution is increased from N96 to N512. The clearest and strongest improvement is seen in autumn for both the location parameter $\mu$ and the scale parameter $\sigma$ (see also Fig. S3). There is also an improvement in $\mu$ in winter and spring, but, for Europe as a whole, biases in $\sigma$ do not decrease with higher resolution in these seasons. The resolution sensitivity in both GEV parameters is small

in summer, with possibly a slightly better performance for the N96 model. Arguably, this result is due to the fact that extreme orographic precipitation in the higher resolution models is larger than that in E-OBS, as also seen for mean precipitation (Fig. 1). The performance of the N216 model is generally in between that of the N96 and N512 models, consistent with the result obtained for mean precipitation (Fig. 2), and also seen in maps of GEV parameter biases analogous to the ones shown in Figures 5 and 6 (not shown).

## 5   Discussion

While a comprehensive analysis of how and why mean and extreme precipitation are sensitive to resolution in HadGEM3-GA3.0 is beyond the scope of this evaluation paper, we briefly discuss two relevant issues, namely the role of the large-scale circulation, specifically that of the North Atlantic storm track, and that of orography, in this section.

### 5.1   North Atlantic stormtrack

European precipitation is strongly determined by the character of the North Atlantic and Mediterranean stormstracks. According to Hawcroft et al. (2012), roughly 70% of European winter precipitation is associated with extratropical cyclones. It is therefore logical to ask to what extent the resolution sensitivity seen in the precipitation is due to resolution sensitivity in the simulation of the stormtrack. By analysing the historical simulations of the models participating in phase five of the Coupled Model Intercomparison Project (CMIP5), Zappa et al. (2013) identified four models with a comparatively good representation

of the North Atlantic stormtrack. These four models have horizontal grid spacings of about 100 km, which is at the high end of CMIP5 model resolutions. The resolution dependence of European precipitation in the EC-EARTH AGCM version 2.3 was analysed by van Haren et al. (2015) by comparing simulations at ≈112 and ≈25km grid spacing, and the authors find that at 25 km EC-EARTH has a better representation of the winter North Atlantic stormtrack and therefore precipitation over Europe.

Similar to Blackmon (1976) and van Haren et al. (2015), we evaluate the representation of the North Atlantic stormtrack

in HadGEM3-GA3.0 by calculating the standard deviation of the 2–8 day band-pass filtered 500hPa geopotential height. The results are shown in Fig. 7 for winter. At all resolutions, the HadGEM3-GA3.0 stormtrack location and strength is similar to that in ERA-Interim (Fig. 7a,b,d,g). The model biases with respect to ERA-Interim (Fig. 7c,f,j) show an overestimation of the Mediterranean storm track strength by about 10%, and an underestimation of about 10% over northeast Europe, though these differences are not significant given the interannual variability. The resolution sensitivity in the stormtrack strength over

Europe attains values of about 5% of the mean (Fig. 7e,h,i). Interestingly, the resolution sensitivity is of opposite sign for the two resolution increases: there is a reduction in storm track strength when going from N96 to N216 over central and western



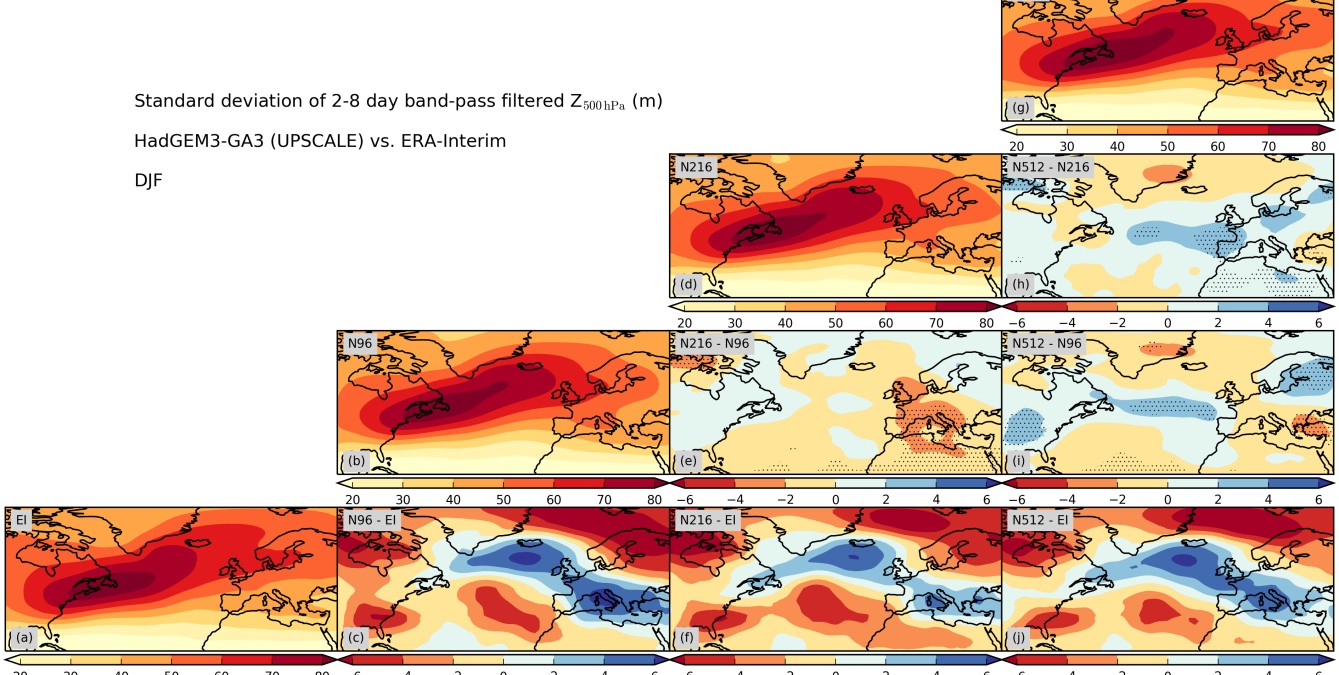

**Figure 7.** Standard deviation of 2–8 day band-pass filtered 500 hPa geopotential height (m). (a) ERA-Interim reanalysis, (b,d,g) model (HadGEM3-GA3) at resolutions N96, N216, and N512, (c,f,j) model biases with respect to ERA-Interim, and (e,h,i) differences between model resolutions. Stippling shows statistically significant differences.

Europe and the Mediterranean (Fig. 7e), but then an increase of similar magnitude over much of Europe when going from N216 to N512 (Fig. 7h).

We proceed by revisiting the resolution sensitivity in mean precipitation, shown in Fig. 1 and in greater detail in Fig. 8, and by comparing it to the sensitivity seen in the stormtrack. In winter, precipitation decreases with resolution over the Northern European Plains and increases over nortwestern and western European coasts, Iberia, and to the south of the Alps and Italy (Fig. 8i). In contrast to the sensitivity in the stormtrack, this pattern can be seen for both steps of resolution increase (Fig. 8e,h) showing that the stormtrack sensitivity is not the main factor explaining the sensitivity seen in mean precipitation. At the same time, the precipitation sensitivity is generally smaller for the N216 to N512 resolution increase than for the N96 to N216 increase, especially so over the Northern European Plains, where the drying with higher resolution is comparatively small (Fig. 8h). This is consistent with increased stormtrack activity in this region when going to N512 resolution (Fig. 7h) suggesting that stormtrack changes with resolution may somewhat modulate the total precipitation sensitivity in HadGEM3-GA3.0.

Similar conclusions can be drawn for summer: the storm-track strength decreases as resolution is increased from N96 to N216 and it increases as resolution is increased from N216 to N512 (Fig. S6e,h). Over the northwest of Europe and the North





Seasonal Mean Precipitation (mm day$^{-1}$)

HadGEM3-GA3 (UPSCALE) vs. E-OBS

DJF

**Figure 8.** Winter (December–February) mean precipitation. (a) Observations (E-OBS), (b,d,g) model (HadGEM3-GA3.0) at resolutions N96, N216, and N512, (c,f,j) model biases with respect to E-OBS, and (e,h,i) differences between model resolutions. Stippling shows statistically significant differences.

Sea there is a slight increase of precipitation when going to N512 resolution (Fig. S7h) consistent with the corresponding increase in storm-track strength (Fig. S6h). In other parts of Europe, however, the precipitation response to resolution is fairly similar for the two steps of resolution increase (compare Fig. S7e with Fig. S7h).

In the transition seasons, too, there is an decrease of storm-track strength for the N96–N216 resolution increase (Figures S4e and S8e), but an increase of storm-track strength for the N216-N512 resolution increase (Figures S4e and S8e). With the



exception of Iberia in spring, no such funamental difference is seen for the resolution sensitivity of mean precipitation (Figures S5e,h and S9e,h).

We have shown here, using a simple metric based on synoptic-scale geopotential-height variance, that mean stormtrack changes with resolution do not primarily explain mean precipitation changes with resolution in our model. More detailed

analyses based on individually tracked extratropical cyclones (Hoskins and Hodges, 2002) and the precipitation associated with them (Hawcroft et al., 2012) are required for a comprehensive assessment of the role of storms in the resolution sensistivity of extreme precipitation in particular. We are currently conducting such analyses in a separate study.

## 5.2   Orography

It has been shown in previous sections that in and around mountainous areas there is particularly high sensitivity to resolution in

both mean (Figures 1 and 8) and extreme (Figures 4–6) precipitation. We therefore investigate the role of orography explicitly in this section. To this end, we conduct a sensitivity experiment where we use HadGEM3-GA6.0 (see Sect. 2.1) at high resolution (N480, i.e. very similar to N512) but apply orographic boundary conditions at coarse (N96) resolution. This sensitivity experiment is similar to the orography experiment in Schiemann et al. (2014), except that in this study we bilinearly interpolate the N96 orography onto the N480 grid to avoid "blocks" of grid boxes of constant orographic height on the N480 grid. The

interpolation is applied both to the (resolved) grid-box mean orographic height and to the boundary conditions used by the parameterisations that represent different effects of subgrid orography. The sensitivity of European mean precipitation to resolution seen in HadGEM3-GA6.0, i.e. for N96–N216–N480 resolution increases, is very similar to that seen in HadGEM3-GA3.0 (not shown).

The results of the orography sensitivity experiment are shown in Fig. 9 for winter. The panels on the diagonal show mean

precipitation for the observations (E-OBS), the N480 experiment with N96 orography (N480$_{N96}$) and the N480 control experiment (N480$_{N480}$), and the off-diagonal panels show the differences between these three fields. These results can be compared to the corresponding results of the full resolution sensitivity experiment, especially the ones corresponding to the N96–N512 resolution increase (Fig. 8). First, comparing N96 with N480$_{N96}$ (Fig. 8a–c and Fig. 9a–c), it can be seen that the precipitation distribution and its bias with respect to E-OBS are broadly similar in these two experiments. Likewise, the total resolution

sensitivity (Fig. 8i) and the effect of introducing high-resolution orographic boundary conditions (Fig 9e) are similar, both in terms of the geographical distribution and the magnitude of the response. This similarity is particularly clear near complex orography such as the Alps and along the Scottish and Norwegian coastlines, but there is also agreement in a large-scale drying with higher resolution (and with more highly resolved orography) over a wide area of the Northern European Plains, and some precipitation increase over the very northeast of Europe. An exception to this overall similarity is the Iberian Peninsula, where

the full response to the resolution increase is a precipitation increase, but N480$_{N480}$ is drier than N480$_{N96}$ over much of the Peninsula. In summer, the total precipitation sensitivity is also very simular to the response to increasing the resolution of orography only (compare Figures S5i and S11e). In summary, these results show that better resolved orography at the higher resolution is a major factor determining the total model sensitivity to resolution in mean precipitation over Europe.



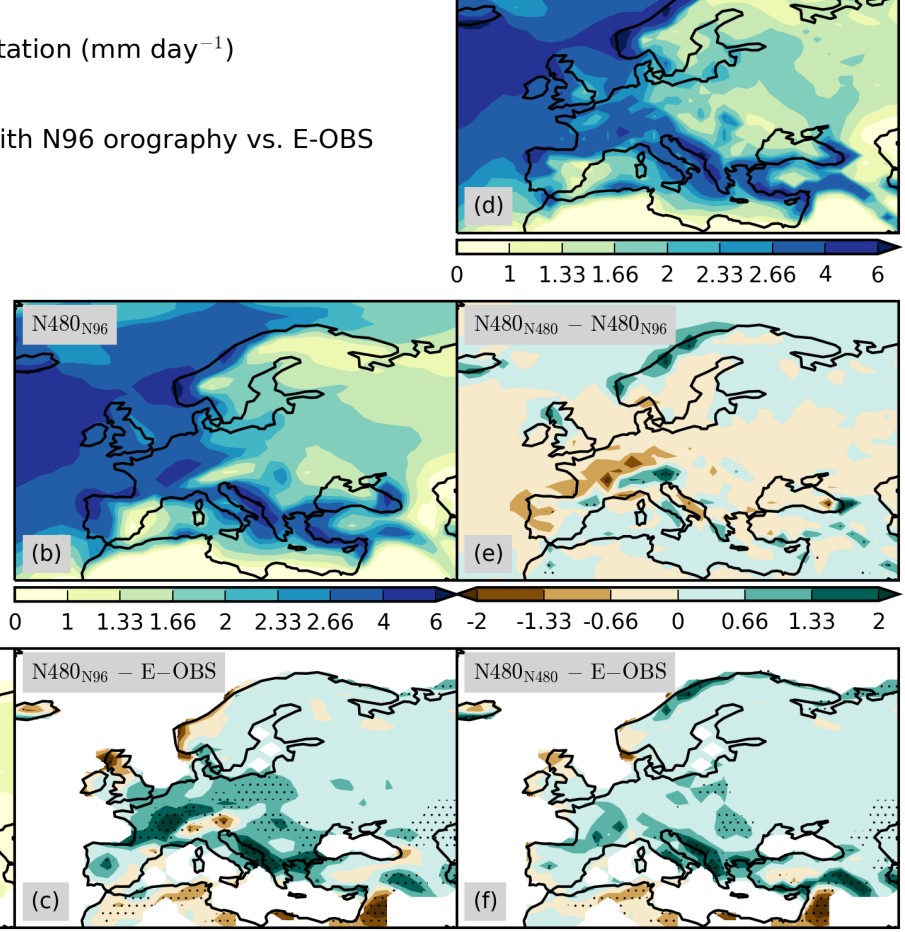

**Figure 9.** Winter (December–February) precipitation in mm day$^{-1}$ in (a) observations (E-OBS), (b) the N480 model with N96 orography, and (d) the N480 control simulation. (c,f) Model bias with respect to E-OBS, (e) difference between the two model simulations. Stippling shows statistically significant differences.

We proceed by comparing the effect of orography with the total sensitivity to resolution for extreme precipitation. The ratios of GEV parameters of the N512 and N96 model for winter are shown in the upper panels of Fig. 10 and should be compared with the same ratios for the N480$_{N480}$ and N480$_{N96}$ models in the lower panels of Fig. 10. Analogous figures for the other seasons are shown in the Supplememt (Figures S12–S14). As seen earlier, the resolution increase leads to an overall reduction

5 of both the location and scale parameters at the N512 resolution (Fig. 10a,c). This reduction is particularly pronounced over the Northern European Plains in northwest and central Europe. At the same time, the GEV parameters increase with resolution for some of the Alpine basins (Rhône, Po, Upper Danube). In this northwest/central part of Europe, a fairly similar pattern can





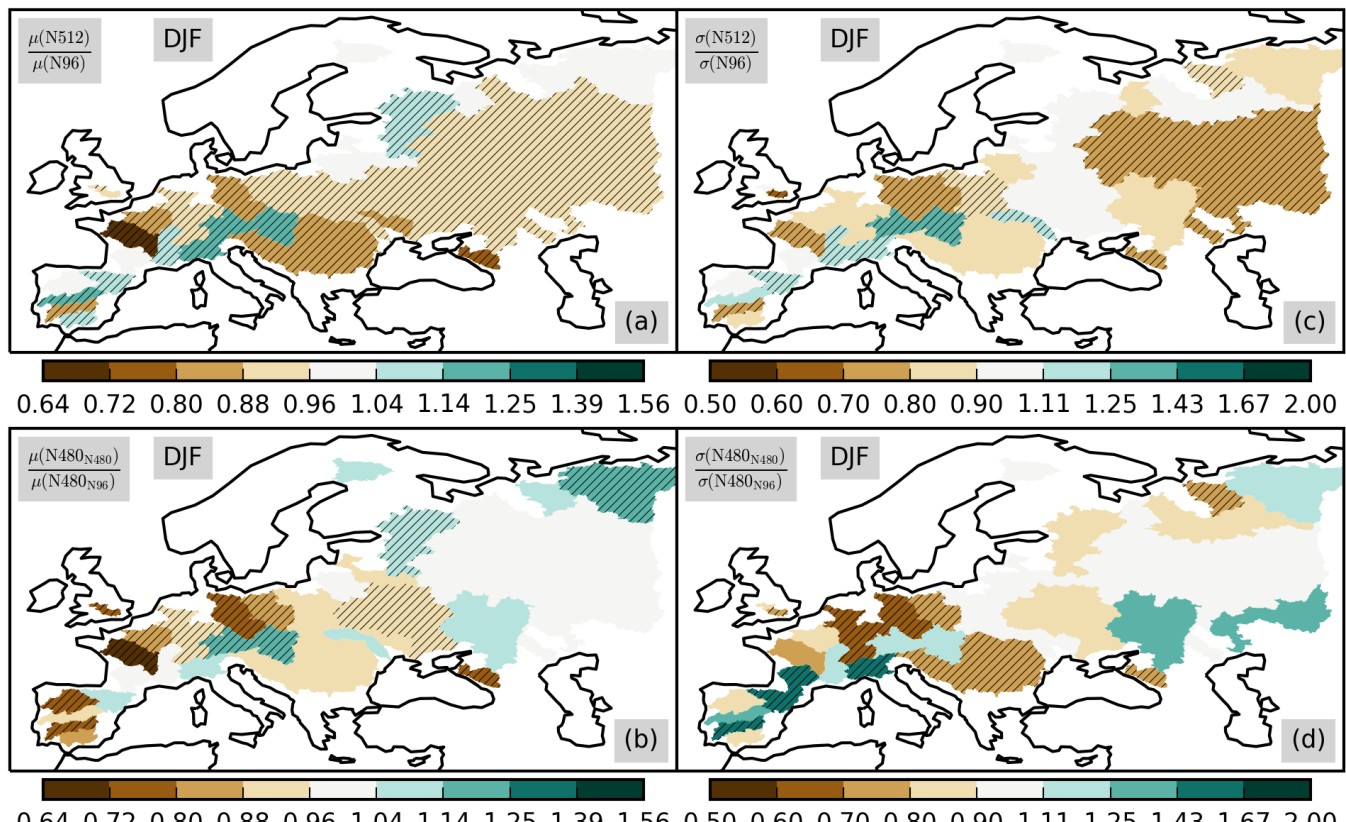

**Figure 10.** Winter (December–February) ratios of fitted GEV parameters for (a,b) the location parameter $\mu$ and for (c,d) the scale parameter $\sigma$, between (a,c) the N512 and N96 simulations and between (b,d) the N480 control simulation and the N480 simulation with N96 orography. Hatching shows statistically significant differences.

be seen when considering the resolution increase in the orographic boundary conditions in isolation (Fig. 10b,d). In other parts of Europe (the Iberian Peninsula, Eastern Europe), there is no clear correspondence between the sensitivity to more highly resolved orography and the total resolution sensitivity. In summer, too, the total and orography-only responses are broadly similar and agree on smaller extremes at higher resolution for most basins (Fig. S14). In summary, orographic effects dominate

5    the sensitivity to resolution in simulated extreme precipitation around the Alps and over the Northern European Plains, but not necessarily in other parts of Europe.

## 6   Conclusions

In this study, we have evaluated the representation of mean and extreme precipitation over Europe in an ensemble of simulations with the HadGEM3-GA3.0 GCM at resolutions between N96 (about 135 km in the midlatitudes) and N512 (about 25 km).

10   This model ensemble has been designed to test the immediate effects of the resolution increase, and re-tuning at the different



resolutions has been kept to a minimum. Convection is parameterised at all three resolutions. We have evaluated HadGEM3-GA3.0 against gridded observations from the E-OBS dataset. For the representation of mean precipitation, we find that:

- The continent-wide mean precipitation in HadGEM3-GA3.0 is greater than that in E-OBS by, depending on the season, 20-50%.

- After correcting for the continent-wide mean bias, the root-mean-squared error in the spatial precipitation field is between 0.5 and 1 mm day$^{-1}$, with larger values during winter than during summer.

- As the resolution is increased from N96 to N512, the model biases in mean precipitation decrease in winter, spring, and autumn. The largest improvement with resolution is seen in winter, when the RMSE is reduced by about 25%. The resolution sensitivity is very small in summer.

- During winter, the spatial bias pattern shows too little precipitation over the very north and west of Europe (Scottish and Scandinavian Mountains), over the Iberian Peninsula, and south of the Alps, and too much precipitation over the rest of Europe, in particular in the Northern European Plains to the north and west of the Alps. These biases are significantly reduced as resolution is increased to N512.

- During summer, the main precipitation bias is a wet bias in the north and northeast of Europe. This bias improves a little with resolution, but at the same time there is a wet bias over areas of high topography (Alps, Scandinavian Mountains) that increases with resolution.

We have evaluated extreme daily precipitation by estimating the extreme value (GEV) distribution location and scale parameters over large (>50000 km$^2$) European river basins in the model simulations at different resolutions and the E-OBS observations. We find that:

- Typical (median) biases in the location and scale parameters are around 20%, but these biases can take very large values, between 50 and 100%, over individual river basins. Most of these very large biases constitute overestimations of the extremes, but underestimations are also seen for a small number of cases in semiarid regions during the warm season.

- Biases in extreme precipitation at these scales are smaller in summer than during the other seasons, around 10% for the median biases in GEV location and scale parameters.

- Extreme precipitation is better simulated as the model resolution is increased. This improvement is seen particularly clearly in autumn, when the median GEV parameter biases for the 25km simulations are less than half of those in the 135km simulations. Improvements are also seen for winter and spring, but not for summer.

We have tested two complementary hypotheses explaining the sensitivity of the simulated mean and extreme precipitation to resolution in HadGEM3-GA3.0. The first hypothesis is that the sensitivity to resolution in precipitation is due to the sensitivity in the large-scale circulation, specifically the North Atlantic stormtrack. The second hypothesis concerns the fact that as the model resolution is increased, the orographic boundary conditions are prescribed at a higher resolution, which impacts the




simulated precipitation. We find that for many areas the improvements in mean winter precipitation seen with increased reso-lution in HadGEM3-GA3.0 are primarily related to the better resolved orography, while resolution sensitivty in the simulated stormtrack plays a lesser role. This important role of orography concerns particularly an improvement seen in the Northern European Plains, from the Loire basin in the west to the Vistula basin in the east. In this area, this result also extends to extreme

precipitation, whereas in other regions, such as Iberia, the better resolved orography is not the main factor explaining resolution sensitivity in extreme precipitation. The dominant role of orography is also seen for summer mean precipitation response to the resolution increase, and broadly also for the resolution sensitivity of extreme precipitation during summer.

In this study, we have quantified biases in mean and extreme precipitation over Europe in a state-of-the-art AGCM, and we have shown that these biases are generally reduced as model resolution is increased. Such biases are important to consider

when assessing if a GCM is suitable for a certain application, and the assessment has to be specific for the particular application at hand. Formal event attribution studies, for example, require very large ensembles of simulations with a model that is able to simulate the extreme value distribution for the type of event under consideration. We have shown that at the scales considered here (daily precipitation and river basins $>50000$ km$^2$), our model exhibits large biases for some of the basins considered. Increasing model resolution to about 25 km may reduce these model biases, but at this resolution the generation of large

ensembles remains computationally prohibitive. For such a challenging application, our results therefore raise questions about whether current global modelling capability is fit for purpose. Numerical downscaling, i.e. nesting a high-resolution RCM in a low-resolution GCM, is a potential alternative, yet at the comparatively coarse resolutions that are currently feasible for the driving GCM, e.g. N96 resolution with 19 vertical levels in the weather@home system (Massey et al., 2015), concerns remain over the ability of the modelling system to represent critically important processes such as midlatitude circulation regimes

(Dawson et al., 2012) or tropical cyclones undergoing extratropical transition (Haarsma et al., 2013). For other applications the assessment will be more positive. For example, small ensembles of simulations with a high-resolution model may yield more credible results for the climate change response in extreme precipitation than simulations with a low resolution model, in situations where the low-resolution model exhibits biases in the simulation of extreme precipitation that are reduced at the high resolution, and where the reasons for the improvement with resolution are sufficiently well understood.

Compared to a study where resolution was increased in a different AGCM (EC-EARTH version 2.3, van Haren et al., 2015), we have shown for HadGEM3-GA3.0 that the sensitivity to resolution seen in the simulated precipitation depends more strongly on the better resolved orography at the higher resolution and that the sensitivity to resolution of the North Atlantic stormtrack is comparatively less important. Our study shows that the role of resolution in different GCMs is not necessarily the same and it is therefore interesting and important to explore the role of resolution systematically in multi-model studies. The

simulations currently carried out within CMIP6-HighResMIP (Haarsma et al., 2016), e.g. in the PRIMAVERA[1] project, will allow for such studies based on a well-designed ensemble of high-resolution coupled GCMs.

---

[1]https://www.primavera-h2020.eu





## 7  Code availability

The MetUM is available for use under licence. A number of research organisations and national meteorological services use the MetUM in collaboration with the Met Office to undertake basic atmospheric process research, produce forecasts, develop the MetUM code and build and evaluate Earth system models. For further information on how to apply for a licence

see http://www.metoffice.gov.uk/research/collaboration/um-partnership. Versions 8.0 (HadGEM3-GA3) and 8.5 (HadGEM3-GA6.0) of the source code are used in this paper. JULES is available under licence free of charge. For further information on how to gain permission to use JULES for research purposes see https://jules-lsm.github.io/access_req/JULES_access.html. Extreme value analysis is based on the R-package gevXgpd developed by Christoph Frei at MeteoSwiss/ETH Zürich.

## 8  Data availability

For access to and documentation of the datasets used in this study see http://www.ecad.eu/E-OBS for the E-OBS precipitation and https://hrcm.ceda.ac.uk for the HadGEM3 simulations.

*Author contributions.*  Contributions by the different authors include conceiving the study, all data analysis and visualisation, and writing of the manuscript (RS), conducting the coarse-orography sensitivity experiment (SJJ, PLV, RS), all aspects of creating the UPSCALE ensemble of simulations (PLV (Principle Investigator), MJR, MSM, RS, MED, JS), and comments on the analysis as it progressed and on the manuscript

(LCS, PLV, MJR, SJJ, MED).

*Acknowledgements.*  RS acknowledges NERC-Met Office JWCRP HRCM funding. PLV, MED, and JS acknowledge NCAS Climate Contract R8/H12/83/001 for the High Resolution Climate Modelling program. PLV (UPSCALE PI) acknowledges the Willis Chair in Climate System Science and Climate Hazards that supports his research. MJR and MSM were supported by the Joint U.K. DECC/DEFRA Met Office Hadley Centre Climate Programme (GA01101). RS, PLV, MJR, and MED also were supported by the PRIMAVERA project under Grant

Agreement No 641727 in the European Commission's Horizon 2020 research programme. LCS received funding from the European Union's Horizon 2020 research and innovation programme under the IMPREX grant agreement No 641811. We thank the team of model developers and infrastructure experts required to conduct the large UPSCALE simulation campaign and acknowledge use of the MONSooN system, a collaborative facility supplied under the JWCRP; the PRACE infrastructure; the Stuttgart HLRS supercomputing center; and the STFC CEDA service for data storage and analysis using the JASMIN platform.



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
