# Peer review of "Mean and extreme precipitation over European river basins better simulated in a 25km AGCM"

_Hydrology and Earth System Sciences, 2017_

## Referee Comment (RC1) · Anonymous Referee #1 · 8 Feb 2018

Review of: " Mean and extreme precipitation over European river basins better simulated in a 25km AGCM"

By: Schiemann et al. Submitted to: Hydrol. Earth Syst. Sci. Manuscript # HESS-2017-732

Recommendation: Minor revisions

Overview:

The Authors investigate the role of the horizontal resolution in representing mean and extreme precipitation over Europe, through a General Circulation Model, with a special

focus on the different results over different river basins.

General comments:

1. This manuscript is well written and seems suitable for publication on HESS after a minor revision. 2. I would suggest to add a chapter, at least some sentences, comparing the 25 km GCM results with state of the art RCM results (EURO-CORDEX) in terms of biases. It is not necessary to show maps but a general evaluation based on existent literature is encouraged. 3. Few minor comment follow.

Specific comments:

- Page 2, line 2: "General Circulation Model", is more appropriate than "Global Climate Model". - Page 3, line 14: Can you expand a bit on the kind of parameterization used? Is there any difference between different resolutions/versions? - Page 4, line 20: in Figure 1,4,5,6 I would see also the intermediate resolution. Also I suggest to plot the bias in b,c,e,f instead of the absolute value. - Page 6, figure 2: I suggest to reduce the y range to 0.4-1.0. - Page 8, line 3: In order to highlight the catchment basin of the three rivers I suggest to use contours (or also the 3 rivers drawn in red colour) in figure 5a. - Figure 4: not clear the meaning of circles: verify the usage of the ""observed maxima" description in figure 4. Also I suggest to uniform the ytick number in all of the subplots. - Figure 7: please add stippling to the model bias. The same for S4 and S6. - Page 15, Discussion: I think that this chapter must also include a discussion relative to chapter 3 and 4 results. - Page 16, line 3: Sentence not true when focusing on the Alps. - Page 20, line 30: I appreciate the "bullets approach" for the Conclusion but this is applied to half of the concluding remarks: the conclusion relative to 5.1 and 5.2 are not listed as bullets. Also part of the bullets is repeated in the last part of the conclusions (page 21).

---

## Referee Comment (RC2) · Anonymous Referee #2 · 15 Feb 2018

General comments

According to the Authors the main aims of the manuscript are:

(1) to provide a benchmark for future generations of GCMs with grid spacings on the order of 10 km, as well as for km-scale RCM simulations with respect to the UPSCALE project atmopsheric GCM (AGCM) simulations in terms of seasonal mean and extreme precipitation;

(2) to provide a methodology (a combination of two previous applied methodologies) to evaluate extreme (daily) precipitation by fitting extreme value distributions and eval-uating the model outputs over large (>50000 skm) river basins in Europe for future

applications as input to impacts models.

(3) to determine to what extent the resolution sensitivity in precipitation is due to the sensitivity to resolution of the simulated North Atlantic storm track and [. . .] to contrast that with the role of local forcing from the orography at different resolutions;

With regard to aims 1 and 2, correctly, the Authors point outs that "is an obvious requirement for climate models to simulate precipitation in a realistic way if the models are to be applied, for example, in process and impact studies of the hydrological cycle [. . .] Due to the wide range of applications, the required realism concerns all aspects of the precipitation distribution in space and time including the probability distribution function of the precipitation time series and its extremes.".

Unfortunately, extreme value analysis results seems to refer to averaged values over selected river basins losing the spatial distribution information that is one of the most relevant for impact studies (together with temperature analysis). The analysis of spatial distribution of extreme precipitation (and their timing) is a point to be investigated/clarified in the manuscript.

A second point to be addressed is the sensitivity of the analysis results to the E-OBS dataset horizontal resolution (not reported in the manuscript), is it possible that the more the AGCM horizontal resolution is close to E-OBS horizontal resolution the better the results are, just because the data spatial resolution is "more similar" and values are not averaged in space ?

With regard to aim 3, the Authors test two alternative hypothesis the sensitivity to "the large-scale circulation, specifically the North Atlantic stormtrack" and to "orography" finding that orography effect is dominant with respect to North Atlantic stormtrack in improving precipitation description using a slightly different AGCM.

Specific Comments Page 10 Line 3-4 Authors write "For most cases, the statistical model fits the observed maxima well, but there are a few discrepancies for larger return

periods of more than about 20 years." This reasonable considering the sample size of 26 years for each ensemble member.

Page 17-Section 5.2 It will be of interest to report how orography varies within the sensitivity experiment, i.e. change in maximum, mean and standard deviation.

Table 1. According to values reported in Tab.1, N216 simulation is the "worst" one, but it is also the same with only 3 ensemble members, of which one is quite different from the other two in winter (Figure 2). Which will be the statistics of N96 and N512 if only 3 member are considered, or, how much does the ensemble size (for a given resolution) affect the statistics of the results? In Figure 2, it is quite evident that N216 winter values are more scattered than N96 and N512 values for the same season.

Technical corrections Page 1 Line 5 the model resolution is indicated as 135km but in other part of the manuscript is 130km, please fix it across the manuscript

Page 2 Line 15 introduce here the meaning of UPSCALE acronym instead of Page 3 Line 6

Final comments I suggest the publication after minor review

---

## Author Comment (AC1) · 9 May 2018

We would like to thank the reviewer for their valuable comments, which we address point by point below. The reviewer's original comments are shown in black and our responses are shown in blue. Line numbers in our responses refer to the revised version of the manuscript submitted herewith.

**General comments**

1. This manuscript is well written and seems suitable for publication on HESS after a minor revision.

   Thank you.

2. I would suggest to add a chapter, at least some sentences, comparing the 25 km GCM results with state of the art RCM results (EURO-CORDEX) in terms of biases. It is not necessary to show maps but a general evaluation based on existent literature is encouraged.

   We would like to thank the reviewer for flagging this omission. In the EURO-CORDEX project, reanalysis-driven RCM ensembles were created at a low (0.44°) and high (0.11°) resolution. While direct quantitative comparisons with our results are difficult due to the precise metrics used, the following qualitative results have been obtained by studies evaluating the role of EURO-CORDEX RCM resolution for simulating European precipitation: For seasonal mean quantities averaged over large European subdomains, no clear benefit of an increased spatial resolution can be identified [Kotlarski et al., 2014]. At the same time, the 0.11° simulations better reproduce spatial precipitation patterns [Casanueva et al., 2016] and this improvement is seen both for mean and moderately heavy (95 and 97.5 daily percentiles) of precipitation [Fantini et al., 2016, Prein et al., 2016]. These benefits due to high resolution are also seen when the evaluation is carried out on the coarser 0.44° grid and are largely attributed to the better representation of orography in the higher-resolution RCMs [Prein et al., 2016]. We have added these previous results to a new paragraph of the introduction of our paper (Page 2, Line 14).

3. Few minor comment follow.

**Specific comments**

4. Page 2, line 2: General Circulation Model, is more appropriate than Global Climate Model.

   The abbreviation is used in both ways. In the context of this paper, the distinction between global and regional models is important so that we refer to our model as a Global Climate Model (GCM).

5. Page 3, line 14: Can you expand a bit on the kind of parameterization used? Is there any difference between different resolutions/versions?

[Figure]

Figure 1: As Fig. 4 of the main manuscript but including the N216 results.

The only difference between the different resolutions consists in a very small number of parameter changes that ensure numerical stability. These changes are described in [Johnson et al., 2016] (Sect. 2.2 and Table 2). We have rephrased one sentence to be more precise, saying that "parameter changes between the different resolutions have been kept to the minimum necessary to ensure numerical stability" instead of "retuning at the different resolutions has been kept to a minimum" (Page 3, Line 21). The model parameterizations are described in [Walters et al., 2011] (Sect. 3).

6. Page 4, line 20: in Figure 1,4,5,6 I would see also the intermediate resolution. Also I suggest to plot the bias in b,c,e,f instead of the absolute value.

For mean precipitation over Europe, the intermediate resolution (N216) results are already included in the main manuscript for winter (Fig. 8) and in the supplement for the other seasons (Figures S5, S7, S9).

A version of Fig. 4 of the main manuscript including the N216 results is shown in Fig. 1. The N216 results are generally between those of N96 and N512 and we have made a corresponding remark in the caption of Fig. 4 of the main manuscript.

We show Figures 5,S2,6,S3 of the main manuscript/supplement but with

the N216 results included in Figures 2–5. Panels b,f,c,g,d,h in these figures do show the model bias and we have adapted the figure captions in the main manuscript/supplement to make this clearer. The N216 results are generally in between those of N96 and N512, and we therefore show N96 and N512 only in the main manuscript/supplement for the sake of brevity, but do include the N216 results in the summary statistics (Tables 2 and 3 of the main manuscript).

7. Page 6, figure 2: I suggest to reduce the y range to 0.4-1.0.

   We find it informative to be able to visually compare the bias reduction due to the resolution increase with the total bias, and therefore prefer to leave the y-axis range at 0–1 mm day$^{-1}$.

8. Page 8, line 3: In order to highlight the catchment basin of the three rivers I suggest to use contours (or also the 3 rivers drawn in red colour) in figure 5a

   This is a good idea and the three basins are now highlighted in Fig 5 of the main manuscript, and this is explained in the captions of both Fig. 4 and Fig. 5 of the main manuscript.

9. Figure 4: not clear the meaning of circles: verify the usage of the "observed maxima" description in figure 4. Also I suggest to uniform the ytick number in all of the subplots.

   The meaning of these circles is described in the caption of Fig. 4 of the main manuscript. We have made this description more precise by referring to the maxima as "block maxima" and saying how many there are in this application (#years $\times$ #ensemble members).

10. Figure 7: please add stippling to the model bias. The same for S4 and S6.

    The model bias with respect to ERA-Interim is not statistically significant and therefore no stippling is seen in Fig. 7c,f,j of the main manuscript. The reason stippling is seen in the model-model differences is because of the larger sample sizes as several ensemble members are available.

11. Page 15, Discussion: I think that this chapter must also include a discussion relative to chapter 3 and 4 results.

    We are not entirely sure what is meant by this comment. Section 5 relates the resolution sensitivity seen in mean and extreme precipitation, described in Sections 3 and 4, to the resolution sensitivity in the storm track and due to the change with resolution of the orographic boundary conditions. Naturally, therefore, these results are referred to in Sect. 5, for example in Fig. 8 of the main manuscript showing mean precipitation biases and resolution sensitivity, in Fig. 10a,c derived from results in Sect. 4, and in the text such as at the start of Sect. 5.2 (Page 16, Line 26). We have added another introductory sentence to Sect. 5 (Page 14, Line 4) to make the role of this section in the paper abundantly clear.

[Figure]

Figure 2: As Fig. 5 of the main manuscript but including the N216 results. Estimated GEV parameters for December-February daily basin-average precipitation, (a-d) location parameter and (e-h) scale parameter, both in mm day$^{-1}$, and for (a,e) observations (E-OBS), (b,f) N96 bias, and (c,g) N216 bias, and (d,h) N512 bias. Stippling (hatching) shows statistically significant biases between the models and E-OBS (differences between N216/N512 and N96).

[Figure]

Figure 3: As Fig. S2 of the supplement but including the N216 results. Estimated GEV parameters for March-May daily basin-average precipitation, (a-d) location parameter and (e-h) scale parameter, both in mm day$^{-1}$, and for (a,e) observations (E-OBS), (b,f) N96 bias, and (c,g) N216 bias, and (d,h) N512 bias. Stippling (hatching) shows statistically significant biases between the models and E-OBS (differences between N216/N512 and N96).

[Figure]

Figure 4: As Fig. 6 of the main manuscript but including the N216 results. Estimated GEV parameters for June-August daily basin-average precipitation, (a-d) location parameter and (e-h) scale parameter, both in mm day$^{-1}$, and for (a,e) observations (E-OBS), (b,f) N96 bias, and (c,g) N216 bias, and (d,h) N512 bias. Stippling (hatching) shows statistically significant biases between the models and E-OBS (differences between N216/N512 and N96As Fig 2 but for June–August.

[Figure]

Figure 5: As Fig. S3 of the supplement but including the N216 results. Estimated GEV parameters for September-November daily basin-average precipitation, (a-d) location parameter and (e-h) scale parameter, both in mm day$^{-1}$, and for (a,e) observations (E-OBS), (b,f) N96 bias, and (c,g) N216 bias, and (d,h) N512 bias. Stippling (hatching) shows statistically significant biases between the models and E-OBS (differences between N216/N512 and N96).

12. Page 16, line 3: Sentence not true when focusing on the Alps.

    It is true that the magnitude is smaller for the N216-N512 resolution increase than for the N96-N216 resolution increase. We have therefore changed our formulation to "... the precipitation response is qualitatively similar ..." (Page 16, Line 15). What is important here, however, is that the sign does not change as it does in the storm track change for N96-N216 compared to N216-N512 (Fig. S6e,h).

13. Page 20, line 30: I appreciate the bullets approach for the Conclusion but this is applied to half of the concluding remarks: the conclusion relative to 5.1 and 5.2 are not listed as bullets. Also part of thebullets is repeated in the last part of the conclusions (page 21).

    This is a good suggestion and we have itemised that part of the discussion as well (Page 21, Line 14).

**References**

[Casanueva et al., 2016] Casanueva, A., Kotlarski, S., Herrera, S., Fernández, J., Gutiérrez, J. M., Boberg, F., Colette, A., Christensen, O. B., Goergen, K., Jacob, D., Keuler, K., Nikulin, G., Teichmann, C., and Vautard, R. (2016). Daily precipitation statistics in a EURO-CORDEX RCM ensemble: added value of raw and bias-corrected high-resolution simulations. *Clim. Dyn.*, 47(3-4):719–737.

[Fantini et al., 2016] Fantini, A., Raffaele, F., Torma, C., Bacer, S., Coppola, E., Giorgi, F., Ahrens, B., Dubois, C., Sanchez, E., and Verdecchia, M. (2016). Assessment of multiple daily precipitation statistics in ERA-Interim driven Med-CORDEX and EURO-CORDEX experiments against high resolution observations. *Clim. Dyn.*, pages 1–24.

[Johnson et al., 2016] Johnson, S. J., Levine, R. C., Turner, A. G., Martin, G. M., Woolnough, S. J., Schiemann, R., Mizielinski, M. S., Roberts, M. J., Vidale, P. L., Demory, M.-E., and Strachan, J. (2016). The resolution sensitivity of the South Asian monsoon and Indo-Pacific in a global 0.35 AGCM. *Clim. Dyn.*, 46(3-4):807–831.

[Kotlarski et al., 2014] Kotlarski, S., Keuler, K., Christensen, O. B., Colette, A., Déqué, M., Gobiet, A., Goergen, K., Jacob, D., Lüthi, D., Van Meijgaard, E., Nikulin, G., Schär, C., Teichmann, C., Vautard, R., Warrach-Sagi, K., and Wulfmeyer, V. (2014). Regional climate modeling on European scales: A joint standard evaluation of the EURO-CORDEX RCM ensemble. *Geosci. Model Dev.*, 7(4):1297–1333.

[Prein et al., 2016] Prein, A. F., Gobiet, A., Truhetz, H., Keuler, K., Goergen, K., Teichmann, C., Fox Maule, C., van Meijgaard, E., Déqué, M., Nikulin,

G., Vautard, R., Colette, A., Kjellström, E., and Jacob, D. (2016). Precipitation in the EURO-CORDEX 0.11 and 0.44 simulations: high resolution, high benefits? *Clim. Dyn.*, 46(1-2):383–412.

[Walters et al., 2011] Walters, D. N., Best, M. J., Bushell, A. C., Copsey, D., Edwards, J. M., Falloon, P. D., Harris, C. M., Lock, A. P., Manners, J. C., Morcrette, C. J., Roberts, M. J., Stratton, R. A., Webster, S., Wilkinson, J. M., Willett, M. R., Boutle, I. A., Earnshaw, P. D., Hill, P. G., MacLachlan, C., Martin, G. M., Moufouma-Okia, W., Palmer, M. D., Petch, J. C., Rooney, G. G., Scaife, A. A., and Williams, K. D. (2011). The Met Office Unified Model Global Atmosphere 3.0/3.1 and JULES Global Land 3.0/3.1 configurations. *Geosci. Model Dev.*, 4(4):919–941.

---

## Author Comment (AC2) · 9 May 2018

We would like to thank the reviewer for their valuable comments, which we address point by point below. The reviewer's original comments are shown in black and our responses are shown in blue. Line numbers in our responses refer to the revised version of the manuscript submitted herewith.

**General comments**

1. According to the Authors the main aims of the manuscript are:

   (1) to provide a benchmark for future generations of GCMs with grid spacings on the order of 10 km, as well as for km-scale RCM simulations with respect to the UPSCALE project atmopsheric GCM (AGCM) simulations in terms of seasonal mean and extreme precipitation;

   (2) to provide a methodology (a combination of two previous applied methodologies) to evaluate extreme (daily) precipitation by fitting extreme value distributions and evaluating the model outputs over large (>50000 skm) river basins in Europe for future applications as input to impacts models.

   (3) to determine to what extent the resolution sensitivity in precipitation is due to the sensitivity to resolution of the simulated North Atlantic storm track and [...] to contrast that with the role of local forcing from the orography at different resolutions;

   With regard to aims 1 and 2, correctly, the Authors point outs that "is an obvious requirement for climate models to simulate precipitation in a realistic way if the models are to be applied, for example, in process and impact studies of the hydrological cycle [...] Due to the wide range of applications, the required realism concerns all aspects of the precipitation distribution in space and time including the probability distribution function of the precipitation time series and its extremes.".

   Unfortunately, extreme value analysis results seems to refer to averaged values over selected river basins losing the spatial distribution information that is one of the most relevant for impact studies (together with temperature analysis). The analysis of spatial distribution of extreme precipitation (and their timing) is a point to be investigated/clarified in the manuscript.

   The choice of river basins of an area of 50000 km$^2$ or more is adequate in view of the fact that the resolution of the GCMs evaluated here is between about 135 km and 25 km. For some kinds of impacts, these scales are relevant as demonstrated for example by flood events affecting much or all of a river basin of this size (e.g. [Grams et al., 2014]). We do, however, fully agree with the reviewer that for other applications smaller scales are important. For such smaller scales, higher-resolution RCM output may be preferable, but the quality of the RCM output may still largely depend on the performance of the driving GCM [Graham et al., 2007]. We have made this point clearer in the introduction (Page 2, Line 33 – Page 3, Line 3). As suggested by Reviewer 1, we have also added an additional

Table 1: Maximum, mean, and standard deviation of orography over European land (-14–50°E) in the control and sensitivity experiments (m).

| | max | mean | std. deviation |
|---|---|---|---|
| N480 | 2844 | 366 | 453 |
| N480$_{N96}$ | 1977 | 337 | 370 |

paragraph summarising results obtained with the EURO-CORDEX RCM ensemble (Page 2, Lines 14–22).

2. A second point to be addressed is the sensitivity of the analysis results to the E-OBS dataset horizontal resolution (not reported in the manuscript), is it possible that the more the AGCM horizontal resolution is close to E-OBS horizontal resolution the better the results are, just because the data spatial resolution is more similar and values are not averaged in space ?

   This is an interesting comment and the reason why we have conducted the scale-dependent evaluation (Fig. 3 of the main manuscript) showing that this is not the case, our results are robust throughout a wide range of spatial scale. As far as extreme precipitation is concerned, this is one of the reasons to aggregate all data sets over river basins larger than 50000 km$^2$, i.e. above the grid scale of any of the models and at a scale where E-OBS is representative (see also previous comment).

   With regard to aim 3, the Authors test two alternative hypothesis the sensitivity to the large-scale circulation, specifically the North Atlantic stormtrack and to orography finding that orography effect is dominant with respect to North Atlantic stormtrack in improving precipitation description using a slightly different AGCM.

**Specific comments**

3. Page 10 Line 3-4 Authors write For most cases, the statistical model fits the observed maxima well, but there are a few discrepancies for larger return periods of more than about 20 years. This reasonable considering the sample size of 26 years for each ensemble member.

   We agree, we cannot sample internal variability on longer timescales than the individual simulations, even with an ensemble.

4. Page 17-Section 5.2 It will be of interest to report how orography varies within the sensitivity experiment, i.e. change in maximum, mean and standard deviation.

   These values are shown in Table 1.

5. Table 1. According to values reported in Tab.1, N216 simulation is the "worst" one, but it is also the same with only 3 ensemble members, of

[Figure]

Figure 1: Alternative versions of Figure 2 in the main manuscript using only three randomly sampled ensemble members for the N96 and N512 models. Three versions of the figure with a different random number generator seed are shown.

which one is quite different from the other two in winter (Figure 2). Which will be the statistics of N96 and N512 if only 3 member are considered, or, how much does the ensemble size (for a given resolution) affect the statistics of the results? In Figure 2, it is quite evident that N216 winter values are more scattered than N96 and N512 values for the same season.

As far as Table 1 of the main manuscript is concerned, the 95% confidence intervals of European mean precipitation tend to be larger for N216 than for N96 and N512. This is indeed due to the fewer ensemble members (3) in N216. The mean values are however very similar for all three resolutions and the confidence intervals strongly overlap, so that there is no clear "worst" model.

The fact that the ensemble size differs (5 members for N96 and N512, and 3 members for N216) does not impact the conclusions drawn from Fig. 2 as can be seen from the results for individual ensemble members included in that figure. To illustrate this further, we show versions of Fig. 2 of the main manuscript but using only three randomly sampled members for the N96 and N512 models (Fig. 1). These figures are very similar to one another and to Fig. 2 in the main manuscript showing that our conclusions are robust.

**Technical comments**

6. Page 1 Line 5 the model resolution is indicated as 135km but in other part of the manuscript is 130km, please fix it across the manuscript

   Thank you for spotting this, we now use 135 km throughout.

7. Page 2 Line 15 introduce here the meaning of UPSCALE acronym instead of Page 3 Line 6

   This has been changed, thank you.

**References**

[Graham et al., 2007] Graham, L. P., Hagemann, S., Jaun, S., and Beniston, M. (2007). On interpreting hydrological change from regional climate models. *Clim. Change*, 81(SUPPL. 1):97–122.

[Grams et al., 2014] Grams, C. M., Binder, H., Pfahl, S., Piaget, N., and Wernli, H. (2014). Atmospheric processes triggering the central European floods in June 2013. *Nat. Hazards Earth Syst. Sci.*, 14(7):1691–1702.